# A Quantum Chemical Deep-Dive into the π-π Interactions of 3-Methylindole and Its Halogenated Derivatives—Towards an Improved Ligand Design and Tryptophan Stacking

**DOI:** 10.3390/ph15080935

**Published:** 2022-07-28

**Authors:** Ruben Van Lommel, Tom Bettens, Thomas M. A. Barlow, Jolien Bertouille, Steven Ballet, Frank De Proft

**Affiliations:** 1Eenheid Algemene Chemie (ALGC), Faculty of Science and Bio-engineering Sciences, Vrije Universiteit Brussel (VUB), Pleinlaan 2, 1050 Brussels, Belgium; tom.bettens@outlook.com; 2Molecular Design and Synthesis, Department of Chemistry, KU Leuven, Celestijnenlaan 200F Leuven Chem&Tech, Box 2404, 3001 Leuven, Belgium; 3Research Group of Organic Chemistry (ORGC), Faculty of Science and Bio-engineering Sciences, Vrije Universiteit Brussel (VUB), Pleinlaan 2, 1050 Brussels, Belgium; thomas.barlow@vub.be (T.M.A.B.); jolien.bertouille@vub.be (J.B.)

**Keywords:** π-π stacking, peptides, density functional theory, indoles, non-covalent interactions

## Abstract

Non-covalent π-π stacking interactions often play a key role in the stability of the secondary and tertiary structures of peptides and proteins, respectively, and can be a means of ensuring the binding of ligands within protein and enzyme binding sites. It is generally accepted that minor structural changes to the aromatic ring, such as substitution, can have a large influence on these interactions. Nevertheless, a thorough understanding of underpinning phenomena guiding these key interactions is still limited. This is especially true for larger aromatic structures. To expand upon this knowledge, elaborate ab initio calculations were performed to investigate the effect of halogenation on the stability of 3-methylindole stacking. 3-Methylindole served as a representation of the tryptophan side chain, and is a frequently used motif in drug design and development. Moreover, an expression is derived that is able to accurately predict the interaction stability of stacked halogenated 3-methylindole dimers as well as halogenated toluene dimers, based on monomer level calculated DFT descriptors. We aim for this expression to provide the field with a straightforward and reliable method to assess the effect of halogenation on the π-π stacking interactions between aromatic scaffolds.

## 1. Introduction

Indole is one of the most abundant heterocycles found in nature, and its importance to biological chemistry can hardly be overstated [1,2,3]. The aromatic structure is found in a wide array of biologically significant natural compounds and processes, ranging from the neurotransmitter serotonin to large complex natural products such as reserpine [4]. In addition, the indole scaffold is a versatile template that is amenable to diverse functionalization, owing to the well-described chemistry surrounding the creation and manipulation of this heterocycle [5]. In 2014, analyses of the FDA’s record revealed the significance of nitrogenous heterocycles in medicinal chemistry, with indole and its derivatives ranking ninth in most frequently observed *N*-containing heterocycles in FDA approved drugs [6]. Examples of highly marketed pharmaceuticals bearing the indole moiety include the anti-emetic *ondansetron*, the tyrosine kinase inhibitor *sunitinib*, and *tadalafil*, used to treat erectile dysfunction (Figure 1). Many of these indole-based small molecule drugs interact with the protein active site through π-π stacking interactions between the drug and one of the natural aromatic amino acids [7,8,9].

Of the natural amino acids, tryptophan (Trp) comprises 1.3% of all ribosomal proteins and peptides [10]. The unique character of its side chain–encompassing the indole ring–allows the amino acid to leverage its hydrophobic and aromatic properties. In contrast to aliphatic amino acids, the indole side chain of Trp enables participation in diverse interactions with other aromatic side chains and plays an essential role in the functional properties of interaction surfaces and ligand binding sites within the protein [11,12,13]. These interactions include classical π-π stacking, but also interactions with formal charges (cation-π or anion-π), as is observed in side chains of arginine and lysine [14,15]. In addition to this, the nitrogen atom present on the indole ring enables hydrogen bonding interactions, which are known to increase the aqueous solubility of the protein [16].

As a consequence of the wide range of non-covalent interactions that indole can take part in, the tryptophan amino acid is also a major determinant of stability and folding behaviour of proteins [17]. This has been elegantly exploited in *tryptophan zipper* (TrpZip) peptides, first developed by Cochran and co-workers [18]. In these dodecameric peptides, two pairs of tryptophans “interdigitate” in a zipper-like fashion, resulting in a highly twisted β-hairpin (Figure 2). Clearly, these relatively weak non-covalent interactions involving the indole side chain can exert enormous influence on the behaviour of bioactive or functional peptides. As a result, subtle changes in the strength of these interactions–such as those induced by substituents on the aromatic ring–can have substantial impact when duplicated many times. Indeed, experimental and computational studies have already shown the considerable impact of substituents on non-covalent interactions involving simple aromatic rings such as benzene or toluene [19,20,21,22].

Modulating the nature and stability of these non-covalent interactions requires a fundamental understanding of the underlying phenomena. Extensive theoretical studies have already been performed to investigate substituent effects in π-π stacking interactions of benzene or toluene derived dimers. Historically, these studies explained substituents effects in π-π stacking interactions to arise from substituent-induced electrostatic changes in the π-aryl system [23,24,25,26]. However, more recent work by Wheeler and Houk has since established that for cases where classical quadrupolar electrostatics fail, local through-space interactions involving the substituent can rationalize the stability trends of the stacked systems [27,28].

While most studies focus on the benzene or toluene dimers, this works aims to expand the existing knowledge on substituent effects on π-π stacking interactions. This is achieved by focussing on the biologically relevant indole ring. More specifically, density functional theory calculations were used to investigate how halogenation of 3-methylindole can influence its stacking stability. Finally, a model is derived that is able to accurately predict the dimer stacking interaction energy, based on (conceptual) density functional theory descriptors calculated at the monomer level.

## 2. Results and Discussion

### 2.1. Locating Stacked Dimers and Revealing the Roots of Their Stability through Energy Decomposition Analyses

In this study, 3-methylindole (labelled as ***H***) is used as a simplified model to investigate the tryptophan (Trp) side chain π-π stacking interactions [29]. To discern the effect of halogenation on these non-covalent interactions, derivatives of ***H*** are considered, where mono-halogenation is introduced on the indole’s benzo sites, being either the 4-, 5-, 6-, or 7-position. By taking into account the 4 classical halogen atoms (***X*** = F, Cl, Br or I), 16 derivatives are devised (labelled as ***aX***, with ***a*** the position for halogenation and ***X*** the type of halogen, Figure 3A). Next, combinations of ***H*** with all 16 ***aX*** monomers, and combinations of the different ***aX*** derivatives with each other, keeping the type of halogen constant, resulted in 57 unique dimers (labelled as ***aX-bX***, with ***a*** and ***b*** denoting the position for halogenation and ***X*** the type of halogen).

Herein, we focus our investigation on parallel displaced stacked conformations, because of their importance in the secondary structure of many Trp-containing peptides [9,30,31]. Although other relevant stacking modes, including the T-shaped edge-to-face stacking, is excluded from the present discussion, the structural properties of the bicyclic indole ring necessitate a thorough conformational screening in order to locate the most stable parallel-displaced conformation. The latter was achieved by optimizing the gas phase geometry of the dimer at the M06/cc-pVTZ(-PP) level of theory [32,33,34], starting from a topology where the two monomers are stacked on top of one another, and subsequently continued with topologies where the rings have been rotated in incremental steps of 30° with respect to each other. Following 12 steps covering the entire 360° rotation window, one of the rings is flipped horizontally and the process is repeated to scan the full conformational space (Figure 3B). It should be noted that the elaborate conformational sampling described above was carried out only for the parent ***H-H*** and the ***H-bF*** and ***aF-bF*** type of dimers. When the halogenation pattern featured the addition of the larger Cl, Br or I atom to one or both of the indoles, geometry optimization was initiated from the corresponding most stable fluorine-containing dimer, where F has been substituted by one of the heavier halogens. After locating the 57 most stable parallel-displaced dimers (Figure 3C)–characterized exclusively by non-imaginary frequencies during a vibrational analysis–their gas phase, counterpoise-corrected interaction energies $\left(E_{int}^{CP}\right)$ were computed at the M06/aug-cc-PVTZ(-PP) level of theory (Table 1) [35]. In this manner, accurate interaction energies were obtained, and the basis set superposition error was accounted for. Importantly, the approach and level of theory described above, has previously shown to be adequate for studying π-π interactions of halogenated toluene dimers [22].

The obtained counterpoise-corrected interaction energies of the different dimers show that halogenation has a modest effect on the stabilization of the stacking interaction. Indeed, the $E_{int}^{CP}$ values span a range of 3.6 kcal·mol^−1^, with the most stabilizing stacking interaction observed for the ***5I-5I*** dimer ($E_{int}^{CP}$ = −9.7 kcal·mol^−1^) and the least stabilizing interaction observed for the ***5F-5F*** dimer ($E_{int}^{CP}$ = −6.1 kcal·mol^−1^). Although one might deduce from these results, that the effect of halogenation on the stacking stability of the Trp moiety is negligible, it is important to realize that the effect is additive. Thus, in larger peptides, where multiple stacking interactions are available, subtle changes in stacking stability can have a substantial effect on the stability of the secondary structure [36]. Upon closer analysis of the results, it is noteworthy that substitution with the heavier halogens generally results in a more stable parallel displaced π-π stacking, in the order of F < Cl < Br < I when comparing dimers with the same substitution pattern (Figure 4A). This effect is even more pronounced when both 3-methylindoles are substituted with a halogen (Figure 4B). That being said, it should be emphasized that the effect of halogenation on the stacking stability is also dependent on the substitution pattern. For example, the **7I-7I** dimer is characterized by a value of $E_{int}^{CP}$ = −8.6 kcal·mol^−1^, while the **5I-5I** dimer is more stabilized having a value of $E_{int}^{CP}$ = −9.7 kcal·mol^−1^. Nevertheless, because of the variable stability observed for dimers that bear different halogens, there seems to be no clear, uniform relationship between the substitution pattern and the stability of the stacked dimer.

To gain a more comprehensive understanding behind the non-covalent π-π stacking interactions, a Ziegler-Rauk energy decomposition analysis (EDA) was performed at the M06/TZ2P level of theory, with relativistic effects accounted for by the ZORA formalism [37,38,39,40]. Crucially, the interaction energies obtained from the EDA $\left(E_{int}^{\;}\right)$ reproduce similar trends as the $E_{int}^{CP}$ interaction energies (Appendix A). Not surprisingly, the EDA pinpoints electrostatics ($V_{elstat}^{\;}$) as the major driving force behind the stabilizing stacking for all 57 dimers (Table 1 + Figure 4C). Nevertheless, to obtain information on the effect of halogenation, it is best to take note on the changes that occur during substitution. Therefore, the different factors resulting from the energy decomposition are also expressed relative to the values obtained for the parent ***H-H*** dimer (Appendix A). When comparing the ***H-4X*** dimers, we can deduce that a more stabilizing interaction in the order of F < Cl < Br < I can be explained through both an increased stabilization of the electrostatic contribution ($V_{elstat}^{\;}$) and orbital interaction ($E_{\mathrm{oi}}^{\;}$) (Figure 4D). However, this is not a uniform observation. Indeed, for example, when comparing the ***4Cl-4Cl*** dimer with the ***4Br-4Br*** dimer, we can see that the latter is characterized by a more stabilizing interaction, driven by a combined decreased destabilizing Pauli repulsion ($E_{Pauli}^{\;}$), as well as an increased stabilizing orbital interaction ($E_{oi}^{\;}$) (Figure 4E). The situation becomes even more unsystematic when trying to study the effect of the substitution pattern (Figure 4F). Clearly, in order to unravel the factors that determine how halogenation affects the stabilization in parallel-displaced 3-methylindole stacked dimers, one needs to investigate the dimers on a case-by-case basis.

### 2.2. Linking Interaction Strength to Monomer Properties

In the next step of our research, inspired by the seminal works of the Wheeler group, we aimed to deduce easily computable properties at the monomer level, or combinations thereof, that are able to accurately project the counterpoise-corrected interaction energy ($E_{int}^{CP\;}$) of the corresponding stacked 3-methylindole dimer [29,41,42]. The advantage of this approach is apparent, as calculations performed at the monomer level are less labour- and computationally intensive compared to calculations on the dimer, due to the increased size and conformational degrees of freedom of the latter. In our screening of potential descriptors, global properties of the monomer are included, such as the electrostatic potential (ESP), the dipole moment (μ), the isotropic polarizability (α), and in the context of conceptual Density Functional Theory (cDFT), the electron affinity (EA), the vertical ionization potential (IE), the chemical hardness (η) and the softness (S) [43]. With previous studies showing that the stability afforded by π-π stacking interactions can be linked to properties calculated in a plane parallel and 3.25 Å above the aromatic scaffold [29], other local descriptors were considered. These include descriptors based on the ESP in this plane (maximum value (ESP_max_), minimum value (ESP_min_) and range (ESP_range_)), the local softness indices for a radical reaction (s^0^), a nucleophilic attack (s^+^) and an electrophilic attack (s^-^) as well as the component of the anisotropic polarizability tensor that is oriented perpendicular to the aromatic ring (α_zz_) (Figure 5). The values of all calculated monomer descriptors are provided in the Appendix A while the method to obtain the descriptors is outlined in the computational details section. Finally, to take into account the properties of both monomers that constitute the dimer, descriptor values that relate to the dimer were taken as the sum over the respective monomer descriptors.
(1)$$E_{int}^{Predict}=-0.02636\underset{monomer}{\sum }\left(\alpha \right)+1.791\underset{monomer}{\sum }\left(s_{max}^{0}\right)+29.501\underset{monomer}{\sum }\left(IE\right)-18.099\;\;\;\;$$

With these properties in hand, a stepwise multivariate linear regression analysis was performed in an attempt to derive a model that is able to accurately predict the computed $E_{int}^{CP\;}$ values based on the monomer level computed descriptors. First, a subset containing 14 dimers was created by random stratification, withholding approximately 25% of the data for validation purposes. Next, the calculated descriptors of the remaining 43 dimers were included as a first order effect to obtain a model through a standard least squares approach. Descriptors that have an insignificant relationship to the model, i.e., descriptors that are characterized by a high p-value during a partial effect test, were eliminated in a stepwise fashion [44]. This ultimately avoids adding nonsensical flexibility to the model (see Appendix A for full details on the data partitioning and model optimization). The resulting actual by predicted plot of the model is presented in Figure 6, together with Equation (1).

We were pleased to observe that the model is able to retrieve the $E_{int}^{CP\;}$ values with an appreciable accuracy (R^2^ = 0.73 and RMSE = 0.38 kcal·mol^−1^ for the training data, Appendix A). Furthermore, the validation statistics also verify the predictive ability of the model and showcase the absence of overfitting related issues (R^2^ = 0.81 for validation data). Through analysis of the model’s equation (Equation (1)), the key role of the isotropic polarizability (α), the local softness descriptors $s_{max}^{0}$ and the vertical ionization energy (IE) becomes clear. Based on the coefficients associated with these descriptors, the most stable parallel displaced interaction energy can be obtained when combining 3-methylindole derived monomers with a high polarizability, but a low $s_{max}^{0}$ and vertical ionization energy.

### 2.3. General Model Applicability–Cross-Halogenation and Phenylalanine Stacking

Thus far, only 3-methylindole dimers bearing a single type of halogen (***aX-bX***) have been investigated. However, a logical expansion can be envisaged through the inclusion of dimers where the monomeric units contain different halogens (***aX-bY***). Inclusion of these dimer types resulted in an additional 79 stacking interactions. However, because of the decreased symmetry of the parallel-displaced ***aX-bY*** type dimers compared to the ***aX-bX*** type dimers, explicitly computing the interaction energies of the most stable parallel displaced stacking for all these dimers, would be a laborious and time-consuming effort. Consequently, we aimed to evaluate the ability of the model, trained on ***aX-bX*** dimers, to predict the interaction strength of an ***aX-bY*** dimer. To assess this, 5 random ***aX-bY*** dimers (***4F-6I***, ***4I-7Cl***, ***5Cl-6F***, ***5Cl-7Br*** and ***6F-6Br***) were selected and had their geometry optimized and interaction energy calculated. These selected examples vary in types of halogen and substitution pattern, spanning a diverse space of the ***aX-bY*** type dimer set. Comparing the explicitly calculated counterpoise-corrected interaction energies ($E_{int}^{CP\;}$) with the predicted interaction energies ($E_{int}^{CP\;})$, neatly showcases the ability of the model to provide accurate interaction energies for ***aX-bY*** type dimers. Indeed, for the selected 5 examples, a low absolute percentage error on the interaction energy is obtained (Table 2). This small test set establishes the ability to extrapolate the model to the wider ***aX-bY*** type 3-methylindole dimers, and thus gain quantitative information on the stability of these dimers with low effort.

Looking to probe the borders of the applicability domain of the model’s Equation (1), we investigated if this model could also predict accurate stacking interaction energies of other halogenated aromatic dimers. Our prior work on halogenated toluene dimers to investigate phenylalanine (Phe) stacking served as an ideal dataset for this purpose [22]. In this previous investigation, the interaction energies of parallel displaced dimers that consist of halogenated toluene derivatives, were calculated using the exact same method as used in this work. In total, the study investigated 37 dimers that contained toluene and/or halogenated derivatives thereof. These dimers varied in substitution pattern (*ortho*, *meta*, *para*) as well as the type of halogen (***X*** = F, Cl, Br or I) (Figure 7).

By comparing the explicitly DFT-computed interaction energies $\left(E_{int}^{CP\;}\right)$ of the halogenated toluene dimers with the interaction energies derived from the model’s Equation (1) $\left(E_{int}^{Predict}\right)$, the general applicability of the model is established (Appendix A). The predicted values show an analogous trend with the explicitly computed interaction energies (R^2^ = 0.58), which indicates the ability of the model to assess the effect of halogenation on the stability of the stacking interaction (Figure 8A). Moreover, the model is able to accurately retrieve the values of the interaction energies, with most of the differences between the DFT-computed and predicted interaction energies being below 1 kcal·mol^−1^ (RMSE = 0.57 kcal·mol^−1^). In previous work, we highlighted the occurrence of local non-classical CH∙∙∙X hydrogen bonding interactions in the ***mCl-mCl*, *mBr-mBr*** and ***mI-mI*** toluene dimers [22]. These non-covalent interactions occur between the halogen atom and the hydrogen of the methyl group located on the opposite ring, and are enabled through an optimal substitution pattern in combination with a larger sized halogen (indeed for the ***mF-mF*** dimer, such non-covalent interaction was not observable). Interestingly, the data points associated with these dimers, highlighted by a blue circle in Figure 8A, appear as outliers. Removal of these dimers dramatically improved the model’s prediction accuracy (R^2^ = 0.73 and RMSE = 0.39 kcal·mol^−1^, Figure 8B). This suggests that the model is able to accurately derive quantitative information on the stability of π-π stacking interactions of halogenated aromatic scaffolds. However, dimers where other important local non-covalent interactions are present, such as for example non-classical CH∙∙∙X hydrogen bonding, seem to fall outside the applicability domain of the model.

## 3. Materials and Methods

### 3.1. Determination of Interaction Energies

Geometry optimizations and subsequent single point energy calculations were performed in the gas phase using the Gaussian16 software (Revision A.03) at the M06/aug-cc-PVTZ//M06/cc-PVTZ level of theory [32,33,34,45]. For the I atom, to take into account relativistic effects, the cc-PVTZ-PP and the aug-cc-PVTZ-PP basis set were used during the geometry optimization and single point energy calculation, respectively [34,46]. To minimize errors related to integration grid, an ultrafine grid level setting was adopted [47]. The basis set superposition error was accounted for by the counterpoise method [35]. The DFT method described above has previously shown to retrieve accurate trends for stacking interaction energies of halogenated toluene dimers, with results comparable to the ones obtained using the high level, explicitly correlated CCSD(T)-F12b method [22]. The Cartesian coordinates of the optimized systems in this study are collected in the Appendix A.

### 3.2. Energy Decomposition Analysis

A Ziegler-Rauk-type energy decomposition analysis (EDA) was performed on the optimized geometries at the M06/TZ2P level of theory, in conjunction with the ZORA formalism to account for relativistic effects [37,38,39,40]. These calculations were performed using the open-source PyFrag module (version 2019) in combination with ADF 2019 [48,49]. For a detailed theoretical background, we refer to some excellent review literature [50,51]. In short, the EDA is a fragment-based approach, where the interaction energy between the fragments; in this case the deformed monomers, are decomposed into physically meaningful terms (Equation (2)).
(2)$$E_{int}=V_{elstat}+E_{Pauli}+E_{oi}\;$$

In the above equation, $V_{elstat}$ represents the classical electrostatic interaction between the fragments, $E_{Pauli}$ the Pauli repulsion, responsible for the steric repulsion between the fragments and $E_{oi}$ the orbital-interaction energy.

### 3.3. Computing the Monomer Descriptors

All descriptors that were computed at the monomer level were calculated at the same level of theory as the geometry optimizations (M06/cc-PVTZ(-PP)) and are provided in Appendix A. Some of the global properties, including the polarizability and dipole moment, can be directly obtained from a vibrational analysis. The electron affinity (EA) and vertical ionization energy (IE), on the other hand, are calculated as outlined in Equations (3) and (4), where $E_{N}$ represents the electronic energy of the *N*-electron molecular system and $E_{N-1}$ and $E_{N+1}$ the electronic energies of the corresponding N−1 and N+1 systems, respectively.
(3)$$EA=E_{N}-E_{N+1}\;$$
(4)$$IE=E_{N-1}-E_{N}\;$$

After taking note of the negative electron affinity values of the 3-methylindole derivatives, indicating that the anion is not stable with respect to the loss of an electron, it was opted to approximate the chemical hardness (*η*) as suggested by Tozer and De Proft (Equation (5)) [52]. In this equation, the problem of negative electron affinity values is circumvented by making use of the energy levels of the LUMO ($\epsilon_{LUMO}$) and HOMO ($\epsilon_{HOMO}$) instead of the computed electron affinity. In turn, the global softness can be derived through Equation (6) [43].
(5)$$\eta =\frac{\epsilon_{LUMO}+\epsilon_{HOMO}}{2}+IE\;$$
(6)$$S=\frac{1}{\eta }\;$$

Local descriptors were calculated in a plane situated exactly 3.25 Å above the ring scaffold (z-axis). On this plane, a grid is defined where grid points are spaced out 0.1 a_0_ (≈ 0.05 Å) in both the x- and y-dimension, and spans the outer edges of the molecule. The local descriptors are than computed at each grid point in this plane (Figure 9). All grid operations were facilitated by the cubegen utility as implemented in Gaussian [45]. Local descriptors included the calculation of the electrostatic potential at these grid points, to derive the maximum and minimum local ESP as well as the range in the defined grid. In addition, the local softness descriptors for a radical reaction (*s*^0^), a nucleophilic attack (*s*^+^) and an electrophilic attack (*s*^−^) are computed at each grid point through Equations (7)−(9).
(7)$$s^{0}\left(r\right)=S\;\cdot f^{0}\left(r\right)$$
(8)$$s^{+}\left(r\right)=S\cdot f^{+}\left(r\right)$$
(9)$$s^{-}\left(r\right)=S\cdot f^{-}\left(r\right)$$

In these equations, f(r) represents the corresponding Fukui function at each grid point, which itself is defined as in Equations (10)–(12), with $\rho_{x}\left(r\right)$ the electron density of the *x* = *N*, *N*−1 or *N*+1 electron system.
(10)$$f^{0}\left(r\right)=1/2((\rho_{N+1}\left(r\right)-\rho_{N-1}\left(r\right))\;$$
(11)$$f^{+}\left(r\right)=\rho_{N+1}\left(r\right)-\rho_{N}\left(r\right)\;$$
(12)$$f^{-}\left(r\right)=\rho_{N}\left(r\right)-\rho_{N-1}\left(r\right)\;$$

## 4. Conclusions

To conclude, an elaborate quantum chemical analysis was performed on the stability of the 3-methylindole π-π stacking mode. The effect of halogenation on the strength of the parallel-displaced non-covalent interaction is elucidated. Density functional theory calculations indicate that halogenation of the indole ring has a moderate effect on the stability of the stacking interaction. This effect becomes more pronounced for the heavier halogen atoms and the stability of the π-π stacking follows the general trend F < Cl < Br < I. Next to the type of halogen, the substitution pattern also influences the stability, although no general substitution pattern can be discerned to be the most optimal for increasing the stacking stability.

As can be expected, energy decomposition analysis revealed electrostatic interactions to be the main driving force behind the stabilizing effect for all investigated dimers. However, to determine the *modus operandi* through which halogenation influences the stacking stability, one needs to investigate the dimers on a case-by-case basis. Indeed, our results indicated that a difference in halogen atom and/or substitution pattern can have a variable influence on the electrostatics, orbital interaction and Pauli repulsion.

Finally, a model was derived that is able to predict accurate interaction energies based on descriptors that can be more easily computed at the monomer level. These conceptually relevant monomer properties include the vertical ionization potential (IE), polarizability (α) and the maximum local softness indices $\left(s_{max}^{0}\right)$, the latter of which is calculated in a grid situated parallel and above the aromatic ring. Importantly, the applicability domain of the model extends over cross-halogenated 3-methylindole dimers as well as halogenated toluene dimers.

Through this study, we provided the field with detailed quantitative insights into the stacking stability of (halogenated) indoles. Furthermore, additionally introduce a useful method to approximate the parallel-displaced stacking interaction stability of other halogenated aromatic dimers.

## Figures and Tables

**Figure 1 pharmaceuticals-15-00935-f001:**
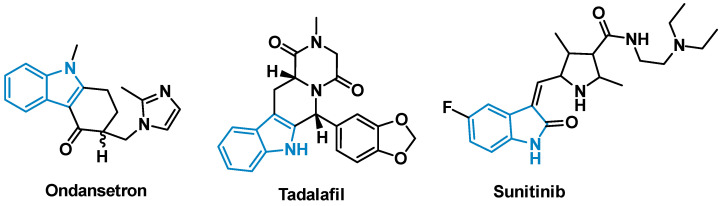
FDA approved indole containing small molecule drugs.

**Figure 2 pharmaceuticals-15-00935-f002:**
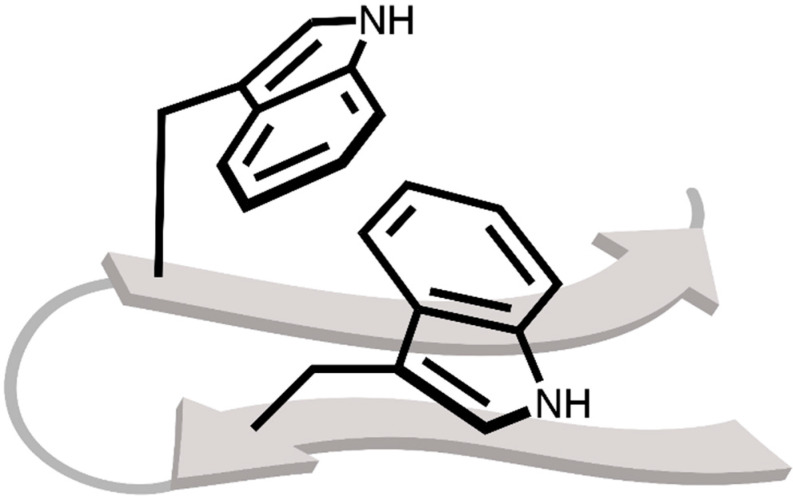
Schematic representation of a TrpZip peptide, only displaying the backbone as a cartoon graphic and the indole side chain bonds. The image focusses in on the indole rings that interdigitate in a zipper-like fashion.

**Figure 3 pharmaceuticals-15-00935-f003:**
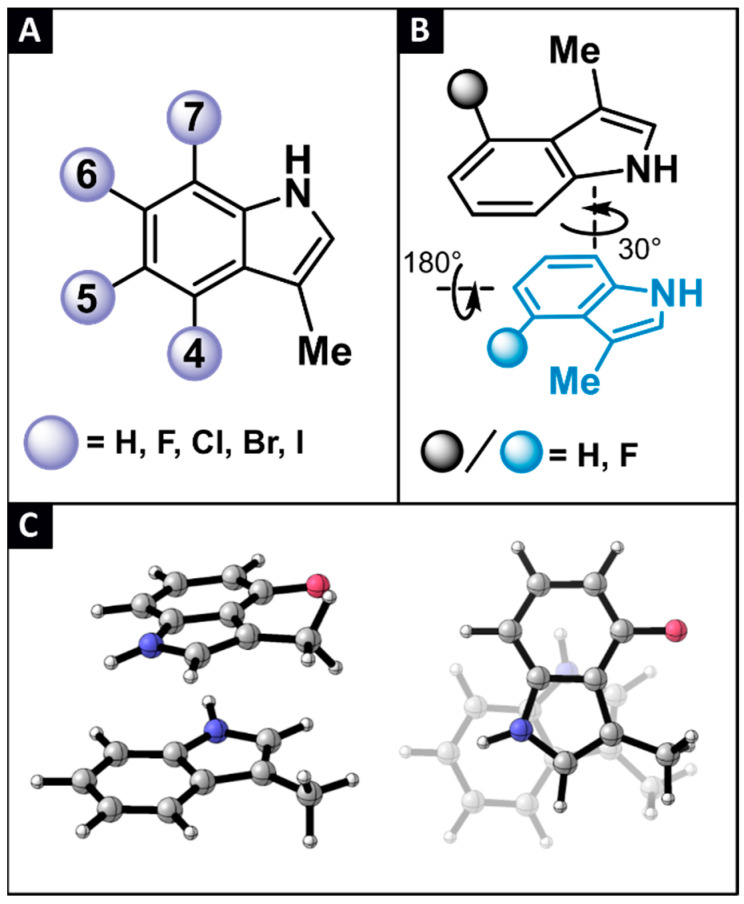
(**A**) A total of 17 monohalogenated 3-methylindole derivatives are considered in this study. (**B**) Schematic representation of the conformational scan performed to locate the global parallel displaced minimum. (**C**) Side- and top-view of the most stable parallel-displaced conformation found for the ***H-4F*** dimer.

**Figure 4 pharmaceuticals-15-00935-f004:**
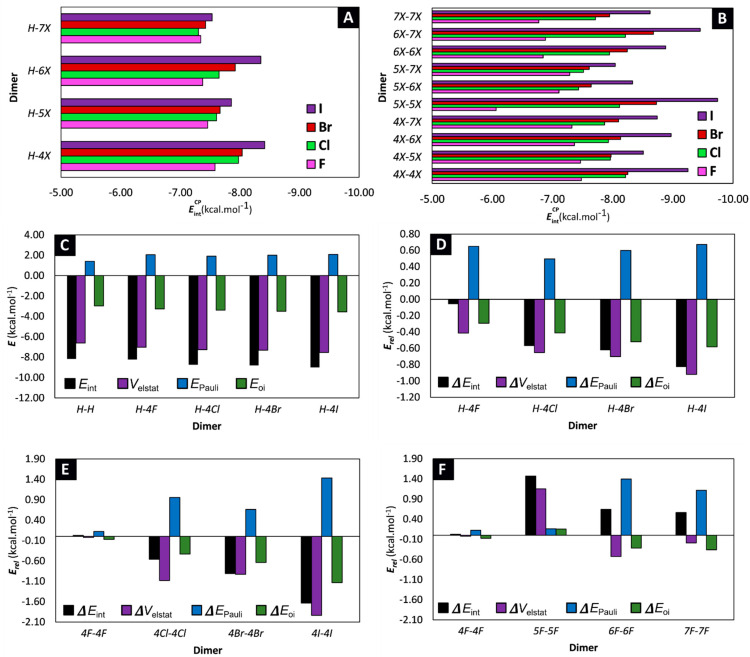
(**A**,**B**) Bar charts of $E_{int}^{CP}$ organized per type of dimer. (**C**) Results of the energy decomposition analysis displayed for the parent **H-H** dimer and **H-4X** dimers. (**D**–**F**) Results of the energy decomposition analysis, expressed relative to the parent **H-H** dimer and displayed for the (**D**) **H-4X**, (**E**) **4X-4X** and (**F**) **aF-bF** dimers. All results are displayed in kcal·mol^−1^, although for clarifying purposes a different scale for the y-axis is used for plots c, d and e and f.

**Figure 5 pharmaceuticals-15-00935-f005:**
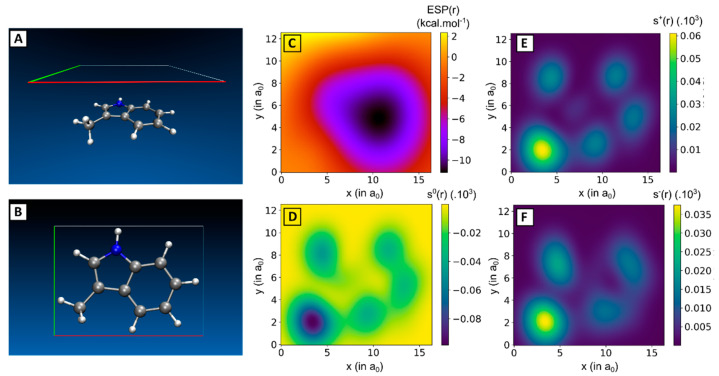
(**A**) Side- and (**B**) top-view of 3-methylindole together with a plane situated parallel and 3.25 Å above the ring scaffold. (**C**) Local electrostatic potential (ESP(**r**)), (**D**) local softness indices for radical attack (s^0^(**r**)), (**E**) nucleophilic attack (s^+^(**r**)) and (**F**) electrophilic attack (s^-^(**r**)) are computed in this plane. Local softness indices values (in a.u.) are multiplied with a factor 10^3^.

**Figure 6 pharmaceuticals-15-00935-f006:**
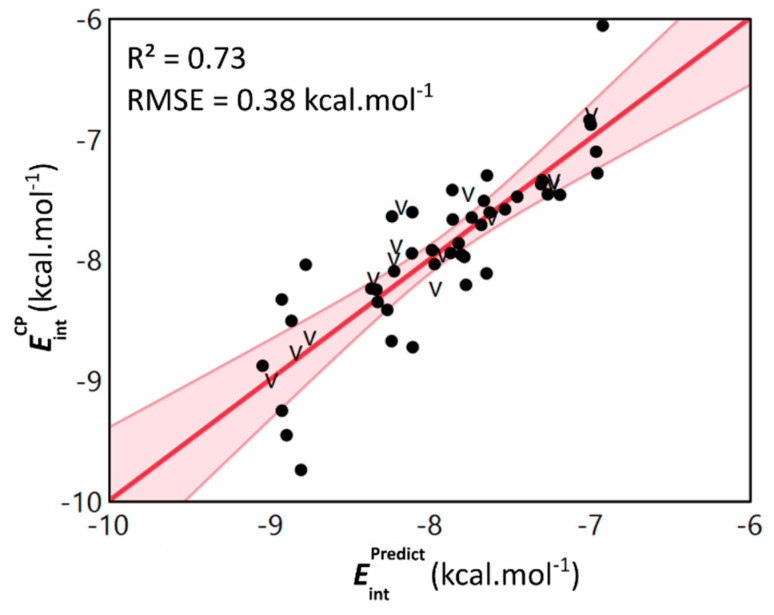
Scatterplot of the dimer level computed counterpoise-corrected interaction energies $E_{int}^{CP\;}$ and the predicted interaction energies $E_{int}^{Predict\;}$ obtained through the monomer level model. Both values are expressed in kcal·mol^−1^ and validation data points are denoted by the letter v. The linear fit and associated confidence interval are shown in red and are accompanied by the R^2^-value and RMSE.

**Figure 7 pharmaceuticals-15-00935-f007:**
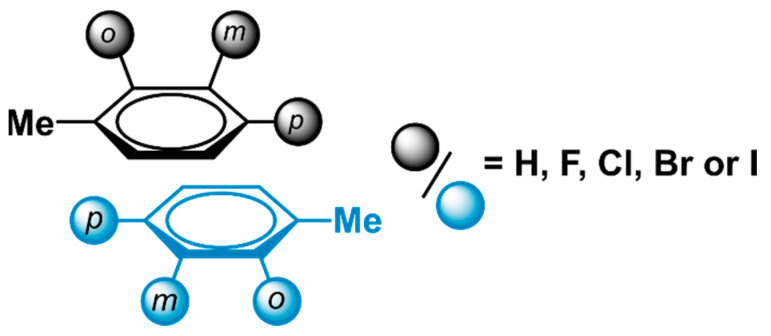
Parallel displaced, halogenated toluene dimers have previously been used to model the stacking interactions observed between phenylalanine residues [22].

**Figure 8 pharmaceuticals-15-00935-f008:**
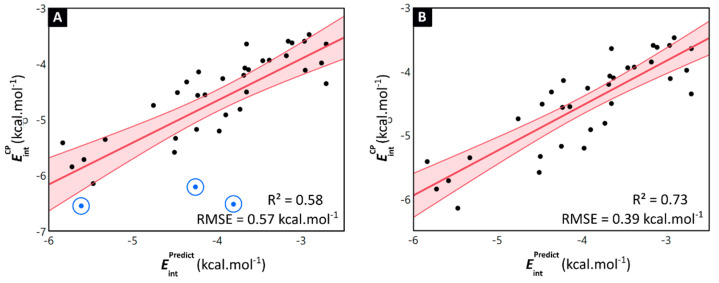
(**A**) Scatterplot of all Phe-dimer level counterpoise-corrected interaction energies$\;E_{int}^{CP\;}$ and the predicted interaction energies $E_{int}^{Predict\;}$ obtained through the monomer level model. Both values are expressed in kcal·mol^−1^. The linear fitting curve and associated confidence interval are shown in red and accompanied by the R^2^-value and RMSE. The ***m*Cl-*m*Cl**, ***m*Br-*m*Br** and ***m*I-*m*I** dimers have been highlighted by a blue circle. (**B**) A remarkable improvement of the fit is obtained when excluding the ***m*Cl-*m*Cl**, ***m*Br-*m*Br** and ***m*I-*m*I** dimers.

**Figure 9 pharmaceuticals-15-00935-f009:**
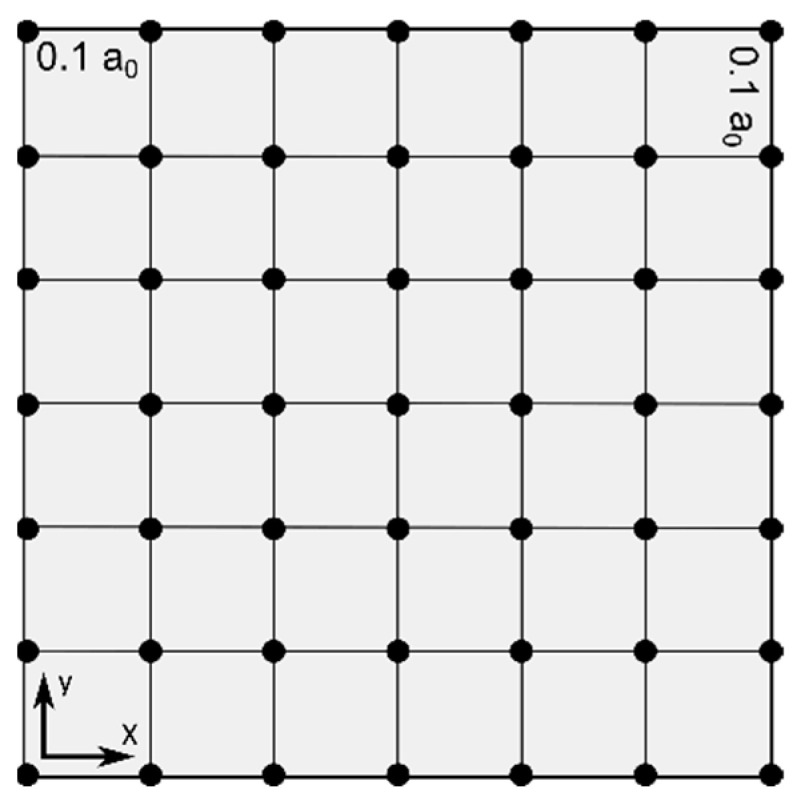
Schematic illustration of grid to compute the local monomer descriptors.

**Table 1 pharmaceuticals-15-00935-t001:** (a) Counterpoise-corrected interaction energies ($E_{int}^{CP})$ are computed at the M06/aug-cc-PVTZ(-PP)//M06/cc-PVTZ(-PP) level of theory. (b) The energy decomposition analysis decomposes the interaction energy ($E_{int}^{\;})$ into the electrostatic ($V_{elstat}^{\;})$, Pauli repulsion ($E_{Pauli}^{\;})$ and orbital interaction ($E_{oi}^{\;})$ terms and has been computed at the M06/TZ2P(ZORA) level of theory. All calculations were performed in the gas phase and values are expressed in kcal·mol^−1^.

**Dimer**	$$E_{\mathrm{int}}^{\mathrm{CP}}\;\;(\mathrm{kcal}\cdot {\mathrm{mol}}^{-1})\;[a]$$	$$E_{\mathrm{int}}^{\;}\;\;(\mathrm{kcal}\cdot {\mathrm{mol}}^{-1})\;[b]$$	$$V_{\mathrm{elstat}}^{\;}\;\;(\mathrm{kcal}\cdot {\mathrm{mol}}^{-1})\;[b]$$	$$E_{\mathrm{Pauli}}^{\;}\;\;(\mathrm{kcal}\cdot {\mathrm{mol}}^{-1})\;[b]$$	$$E_{\mathrm{oi}}^{\;}\;\;(\mathrm{kcal}\cdot {\mathrm{mol}}^{-1})\;[b]$$
*H-H*	−7.6	−8.2	−6.6	1.4	−3.0
*H-4F*	−7.6	−8.2	−7.0	2.1	−3.3
*H-4Cl*	−8.0	−8.8	−7.3	1.9	−3.4
*H-4Br*	−8.0	−8.8	−7.3	2.0	−3.5
*H-4I*	−8.4	−9.0	−7.5	2.1	−3.6
*H-5F*	−7.5	−8.0	−6.6	1.7	−3.0
*H-5Cl*	−7.6	−8.2	−6.7	1.5	−3.0
*H-5Br*	−7.7	−8.3	−6.9	1.8	−3.2
*H-5I*	−7.9	−8.2	−7.0	1.9	−3.0
*H-6F*	−7.4	−7.9	−6.9	2.2	−3.2
*H-6Cl*	−7.7	−8.3	−7.4	2.4	−3.3
*H-6Br*	−7.9	−8.6	−7.6	2.5	−3.5
*H-6I*	−8.4	−8.8	−7.6	2.2	−3.4
*H-7F*	−7.3	−8.0	−6.6	1.7	−3.1
*H-7Cl*	−7.3	−7.9	−6.7	1.9	−3.1
*H-7Br*	−7.4	−8.0	−6.4	1.4	−3.1
*H-7I*	−7.5	−8.0	−6.3	1.3	−3.0
*4F-4F*	−7.5	−8.2	−6.6	1.5	−3.1
*4Cl-4Cl*	−8.2	−8.7	−7.7	2.4	−3.4
*4Br-4Br*	−8.2	−9.1	−7.6	2.1	−3.6
*4I-4I*	−9.3	−9.8	−8.6	2.8	−4.1
*4F-5F*	−7.5	−8.0	−6.8	2.0	−3.2
*4Cl-5Cl*	−8.0	−8.7	−7.7	2.4	−3.4
*4Br-5Br*	−8.0	−8.8	−7.7	2.6	−3.6
*4I-5I*	−8.5	−9.0	−8.2	3.0	−3.8
*4F-6F*	−7.4	−7.9	−6.8	2.0	−3.2
*4Cl-6Cl*	−7.9	−8.7	−7.8	2.6	−3.5
*4Br-6Br*	−8.1	−8.9	−7.8	2.6	−3.7
*4I-6I*	−9.0	−9.2	−8.3	2.7	−3.6
*4F-7F*	−7.3	−8.0	−7.7	3.4	−3.7
*4Cl-7Cl*	−7.9	−8.5	−7.2	1.9	−3.2
*4Br-7Br*	−8.1	−8.8	−7.3	1.9	−3.4
*4I-7I*	−8.7	−8.9	−7.3	1.7	−3.3
*5F-5F*	−6.1	−6.7	−5.5	1.6	−2.8
*5Cl-5Cl*	−8.1	−8.8	−6.6	1.0	−3.2
*5Br-5Br*	−8.7	−9.5	−6.8	0.8	−3.6
*5I-5I*	−9.7	−10.0	−7.3	1.0	−3.7
*5F-6F*	−7.1	−7.4	−6.2	1.7	−2.9
*5Cl-6Cl*	−7.4	−8.0	−7.0	2.1	−3.1
*5Br-6Br*	−7.6	−8.5	−7.4	2.4	−3.5
*5I-6I*	−8.3	−8.6	−7.4	2.1	−3.2
*5F-7F*	−7.3	−7.9	−6.6	1.9	−3.1
*5Cl-7Cl*	−7.5	−8.1	−6.5	1.5	−3.0
*5Br-7Br*	−7.6	−8.2	−6.5	1.4	−3.1
*5I-7I*	−8.0	−8.3	−6.4	1.2	−3.1
*6F-6F*	−6.8	−7.5	−7.1	2.8	−3.3
*6Cl-6Cl*	−7.9	−8.6	−7.3	2.1	−3.3
*6Br-6Br*	−8.2	−9.1	−8.0	2.8	−3.9
*6I-6I*	−8.9	−9.2	−7.7	2.1	−3.5
*6F-7F*	−6.9	−7.7	−7.2	2.8	−3.4
*6Cl-7Cl*	−8.2	−9.0	−8.1	2.9	−3.8
*6Br-7Br*	−8.7	−9.6	−8.3	2.9	−4.1
*6I-7I*	−9.5	−9.8	−8.4	2.7	−4.1
*7F-7F*	−6.8	−7.6	−6.8	2.5	−3.3
*7Cl-7Cl*	−7.7	−8.3	−6.8	1.9	−3.4
*7Br-7Br*	−7.9	−8.6	−6.5	1.5	−3.5
*7I-7I*	−8.6	−8.6	−6.9	1.6	−3.3

**Table 2 pharmaceuticals-15-00935-t002:** Counterpoise-corrected interaction energies ($E_{int}^{CP})$ are computed at the M06/aug-cc-PVTZ(-PP)//M06/cc-PVTZ(-PP) level of theory and predicted interaction energies ($E_{int}^{Predict})$ of the selected dimers. The absolute percentage error of each prediction ($\left|\Delta E_{\;}^{\;}\right|\%)$ is provided as well.

Dimer	$$E_{\mathrm{int}}^{\mathrm{CP}}\;\;(\mathrm{kcal}\cdot {\mathrm{mol}}^{-1})$$	$$E_{\mathrm{int}}^{Predict}\;\;(\mathrm{kcal}\cdot {\mathrm{mol}}^{-1})$$	$$\left|\Delta E_{\;}^{\;}\right|\%$$
*4F-6I*	−8.8	−8.3	6.36%
*4I-7Cl*	−8.4	−8.3	1.43%
*5Cl-6F*	−7.2	−7.3	1.10%
*5Cl-7Br*	−7.6	−7.9	3.28%
*6F-6Br*	−8.1	−7.7	5.42%

## Data Availability

Data is contained within the article and Appendix A.

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
