# Peer review of "A Quantum Chemical Deep-Dive into the π-π Interactions of 3-Methylindole and Its Halogenated Derivatives—Towards an Improved Ligand Design and Tryptophan Stacking"

_pharmaceuticals, 2022, doi:10.3390/ph15080935_

Round 1
Reviewer 1 Report
In this work, the authors made some theoretical calculations to investigate the effect of halogenation of 3-methylindole on π-π stacking interactions. The manuscript is written correctly, the methods are well-chosen, and the results are clearly presented. This study is interesting. The obtained results may be helpful in designing compounds forming stable complexes with proteins. All studied aspects are described step by step in a clear way. I have no negative comments. Only minor editorial corrections are needed:
Line 195, 245 are the same: “Linking interaction strength to monomer properties” – line 245 dimers?
Table 2: different types of fonts in columns 2,3, 4
Line 345 “ …adopted.[47] The…” should be” …adopted [47]. The…” The same mistake on line 232, ref. 44
Eq 5. – something is wrong (line 383, 384)
Author Response
Firstly, we would like to express our appreciation for the positive feedback provided by referee 1. Secondly, we would also like to thank the referee for noticing the editorial flaws that escaped our attention. Accordingly, all suggested corrections by the referee have now been implemented in the manuscript.
Reviewer 2 Report
In this paper, the authors present a computational study of pi-pi stacking interactions in substituted (halogenated) indole dimers. They only considered the most stable parallel displaced stacked conformations (edge to face and parallel eclipsed are not considered). Further, they developed a simple model for predicting interaction energies based on density functional theory descriptors calculated at the monomer level. The model proved well in predicting pi-pi stacking interactions of considered dimers.
The paper is well written, and the results are comprehensively discussed. The only issue I have here is that the interaction energies are rounded to two decimal places, and the accuracy of the applied DFT model is just not enough to justify this. Further, though the model is trained for calculating parallel eclipsed dimers, it would be nice to see whether it works on T-shaped dimers (how much it fails in predicting interaction energies).
Nevertheless, the paper deserves to be published in Pharmaceuticals.
Author Response
We enjoyed reading the referee report from reviewer 2. The reviewer indeed raises some valid points. Firstly, we have decreased the number of decimal places to one, agreeing with the reviewer that the use of two decimal places does not match the accuracy obtained with the applied DFT model. Secondly, we also agree that it would be interesting to see how the model performs in predicting the interaction energy of T-shaped dimers. However, to evaluate this in a statistical relevant manner, one would need to perform a full conformational scan of T-shaped dimers for a number of substituted 3-methylindole dimers. We feel this extensive work falls outside the scope of the presented manuscript and could be a focus of follow-up works that might also include different types of substitution (instead of being confined to halogenation), as well as other biologically relevant arene rings.